# Risk Stratification Model for Severe COVID-19 Disease: A Retrospective Cohort Study

**DOI:** 10.3390/biomedicines11030767

**Published:** 2023-03-02

**Authors:** Miri Mizrahi Reuveni, Jennifer Kertes, Shirley Shapiro Ben David, Arnon Shahar, Naama Shamir-Stein, Keren Rosen, Ori Liran, Mattan Bar-Yishay, Limor Adler

**Affiliations:** 1Health Division, Maccabi Healthcare Services, Tel Aviv 6812509, Israel; 2Department of Family Medicine, Sackler Faculty of Medicine, Tel Aviv University, Tel Aviv 6997801, Israel

**Keywords:** COVID-19, risk stratification model, mortality, hospitalization

## Abstract

Background: Risk stratification models have been developed to identify patients that are at a higher risk of COVID-19 infection and severe illness. Objectives To develop and implement a scoring tool to identify COVID-19 patients that are at risk for severe illness during the Omicron wave. Methods: This is a retrospective cohort study that was conducted in Israel’s second-largest healthcare maintenance organization. All patients with a new episode of COVID-19 between 26 November 2021 and 18 January 2022 were included. A model was developed to predict severe illness (COVID-19-related hospitalization or death) based on one-third of the study population (the train group). The model was then applied to the remaining two-thirds of the study population (the test group). Risk score sensitivity, specificity, and positive predictive value rates, and receiver operating characteristics (ROC) were calculated to describe the performance of the model. Results: A total of 409,693 patients were diagnosed with COVID-19 over the two-month study period, of which 0.4% had severe illness. Factors that were associated with severe disease were age (age > 75, OR-70.4, 95% confidence interval [CI] 42.8–115.9), immunosuppression (OR-4.8, 95% CI 3.4–6.7), and pregnancy (5 months or more, OR-82.9, 95% CI 53–129.6). Factors that were associated with a reduced risk for severe disease were vaccination status (patients vaccinated in the previous six months OR-0.6, 95% CI 0.4–0.8) and a prior episode of COVID-19 (OR-0.3, 95% CI 0.2–0.5). According to the model, patients who were in the 10th percentile of the risk severity score were considered at an increased risk for severe disease. The model accuracy was 88.7%. Conclusions: This model has allowed us to prioritize patients requiring closer follow-up by their physicians and outreach services, as well as identify those that are most likely to benefit from anti-viral treatment during the fifth wave of infection in Israel, dominated by the Omicron variant.

## 1. Introduction

The COVID-19 pandemic has substantially affected human society, causing a significant burden on healthcare systems worldwide [1]. Each wave of the disease has had its unique features, posing different challenges. In Israel, the fifth wave has created a significant challenge regarding resource allocation. This wave was caused primarily by the Omicron variant, first detected in Israel on 26 November 2021. The Omicron variant is known for having a large number of mutations, leading to immune evasion, high transmissibility, and low pathogenicity [2]. During the fifth wave, large numbers of the population were infected daily. The majority of those infected exhibited mild symptoms. However, given the rapid and high infection rates, there was an influx of patients in both the primary care and hospital settings [3]. While new medications for reducing the risk of serious illness had become available at that time, those were not available in sufficient quantities, and there was an urgent need to identify high-risk patients who would benefit most from various treatment options [4].

The process of classifying patients by their risk level is called risk stratification. The Center for Disease Control and Prevention (CDC) identified demographic factors that were associated with a high risk for severe COVID-19 illness, such as older age and specific medical conditions, including chronic kidney failure, diabetes mellitus, and pregnancy [5]. Risk stratification models have been developed to identify those that are at higher risk of COVID-19 infection and severe illness with commonly used characteristics in these models, including vital signs, age, sex, comorbidities, and image features [6]. At the beginning of the pandemic, Barda et al. developed a tool for predicting mortality [7]. However, this was when no vaccinations existed, and other more virulent variants were predominant. Later, models were introduced to prioritize the vaccination effort [8,9]. Hippisley-Cox et al. developed a risk-predicting model for hospitalization or mortality due to COVID-19 [10]. In their model, they accounted for the vaccination status (at this time, only two doses were available). They showed that the absolute risk for hospitalization and mortality was reduced after the second dose. Israel et al. also developed a scoring model including vaccination status. They already included four doses of vaccine [11] and evaluated similar variables as our study did. However, they only addressed the number of vaccines and not their timing and did not consider prior episodes of infection. 

The objective of the present study was to design and develop a scoring tool to identify COVID-19 patients that are at risk for severe illness (hospitalization or mortality). This tool was designed when patients received four vaccine doses and potentially had a prior infection with COVID-19. The novelty of this model is that we considered the number of vaccine doses and their timing. In addition, prior infection was taken into account. This model aimed to assist Maccabi Healthcare Services (MHS), the second largest health maintenance organization (HMO) in Israel, to allocate its resources more effectively by proactively monitoring those at that are risk and providing those that are amongst this high-risk group treatments to reduce the risk of serious illness.

## 2. Methods

### 2.1. Study Design and Setting

We conducted a retrospective cohort study among MHS members in Israel. Under the Israel National Health Insurance law, every Israeli citizen is entitled to various health services provided by the HMO of their choice (four in total). MHS provides healthcare coverage for over a quarter (2.6 million) of the country’s population. The present study is based on data that were extracted from the MHS database. The database includes demographic data for all members and complete records for all outpatient and community-based physician visits, procedures, hospitalizations, prescription purchases, and laboratory tests. MHS has established disease registries based on these data, including heart disease, diabetes, hypertension (HTN), cancer, complicated medical conditions (thalassemia, Gaucher, dialysis, hemophilia, AIDS), chronic obstructive pulmonary disease (COPD), chronic kidney disease (CKD), and obesity (based on BMI) and immunosuppression registries. MHS also maintains a pregnancy status registry based on OB/GYN visit records that include the date of last menstruation and the date of abortion/delivery. The MHS database contains the above mentions parameters for all patients. Therefore, there was no need for data pre-processing and cleaning.

The study population was comprised of all members with a new infection, corroborated by a positive PCR or antigen result between 26 November 2021 (the start of the Omicron wave of COVID-19 infection) and 18 January 2022 (ten days before the end of the study period). The results were verified by using a 2-step approach. The model was first built on the ‘train group’ (one-third of the study population) and later tested and verified on the ‘test group’ (the remaining two-thirds of the study population). Bias was handled by using multivariate regressions. 

### 2.2. Study Database

MHS maintains a COVID-19 registry based on PCR and antigen testing results that are carried out by the MHS Central laboratory and augmented by Ministry of Health updates. The registry includes the date of the first positive result for each new episode of illness and the date of recuperation. New episodes are defined as a positive test result at least three months from the last positive test. The recuperation date is five to ten days from the first positive test result in an episode for low-risk patients/non-hospitalized patients. Higher-risk patients (based on age and medical condition) are followed up by the MHS COVID outreach program and deemed recuperated only after symptoms have cleared. COVID-19 patients that are discharged from the hospital are also followed up by the outreach team and deemed recuperated once asymptomatic. Data were also collected for all members regarding the date, type, and number of COVID-19 vaccinations each member had received. 

### 2.3. Outcomes 

The outcome of interest was either COVID-19-related hospitalization or all-cause mortality, herein referred to as ‘severe illness’, between 26 November 2021 and 27 January 2022. COVID-19-related hospitalizations were defined as any COVID-19-related hospitalization that was reported by the Ministry of Health, hospitalization in a designated COVID-care hospital department, or hospitalization (any department) between the date of diagnosis (first positive test result date for the episode) and date of recuperation. COVID-19-related mortality included all Ministry of Health COVID-19-related reported deaths or member deaths that were recorded between the date of diagnosis and date of ‘recuperation’.

### 2.4. Model Development

A third of the study population was randomly selected to build the scoring model, referred to as the ‘train group’. Factors that were included in the model were gender, age group, socioeconomic status (SES; based on census and national survey classifications applied to home address), nursing home status (based on facilities provided coordinated, on-site medication and vaccination services by MHS), pregnancy status, COVID-19 vaccination status, prior episodes of infection with SARS-CoV-2 (yes/no), and registry status for each of the illness registries that were described above. For those illness registries with inherent high overlap, patients in the immunosuppression registry (e.g., taking immunosuppressant medications/on dialysis) were no longer indicated as included in the complicated medical conditions, CKD, or cancer registries; those patients both in the cancer and complicated medical condition registries were allocated to the complicated medical condition registries only. Pregnancy status was categorized as five or months pregnant (yes/no), as initial analyses indicated that women in their first months were not at greater risk. Vaccine status, as of the date of the first positive result in the episode, was categorized in the following manner: never vaccinated, vaccinated (irrespective of dose number) at least six months ago or more, vaccinated with one to two doses in the last six months, and vaccinated with three to four doses in the last six months.

Univariate analyses between each factor and severe illness outcome were evaluated using Chi-Square analysis. All factors that were associated with severe illness outcomes were then entered into a logistic regression model, β coefficients were calculated for each category, and individual risk probabilities were applied to each participant in the ‘train’ group. Probabilities were then ranked into ten equal groups to determine a risk score between one and ten. The proportion of severely ill patients for each score was calculated, and a dichotomous variable of high-risk for serious illness was developed based on the findings.

### 2.5. Model Evaluation

Risk scores were calculated for the remaining two-thirds of the study population (herein referred to as the ‘test’ group) based on the β coefficients that were developed in the model. Risk score sensitivity, specificity, and positive predictive value (PPV) rates were calculated using the high-risk measure as the outcome of interest. Receiver operating characteristics (ROC) were also calculated to describe the performance of the model in predicting severe illness. Individual risk scores for all MHS members were then calculated, and the high-risk population was updated fortnightly.

All analyses were carried out using SPSS 24 (IBM©, Chicago, IL, USA). The study was approved by the MHS internal review board and the Maccabi Helsinki committee (0052-22-MHS).

## 3. Results

### 3.1. Study Population

Over the two-month study period, just under 410,000 HMO members were diagnosed with COVID-19. No demographic and health characteristics differences were found between the randomized ‘train’ and ‘test’ groups. Descriptive statistics of the study population (both groups combined) are presented in Table 1. Nearly a third of the study population was under 18, and just over 30% were from a high socioeconomic bracket (Table 1). Half the study population was fully vaccinated (Table 1).

### 3.2. Model Components

Of the total study population, 0.4% were either hospitalized or died, with no difference in illness severity rates between the ‘train’ and ‘test’ groups. Factors that were identified in the ‘train’ group that were associated with severe COVID-19 illness were age, vaccination status, prior episodes of COVID-19, immunosuppression status, and pregnancy (Table 2). Age significantly increased the risk of severe illness, with the odds of being severely ill increasing from 45 and dramatically increasing for those aged 75 and over (Table 2). Although a very small group, the risk of severe illness was very high for women in their fifth month or more of pregnancy (Table 2). Of the ‘train’ group, 1,812 women were found positive for COVID-19 while pregnant. None of these women died, but 50 (2.8%) were hospitalized. Most of those hospitalized (88%) were in their fifth month of pregnancy or more. Immunosuppressed COVID-19 patients were also much more at risk of severe illness.

The model identified two protective factors: vaccination status and prior episodes of COVID-19 illness. COVID-19 patients that had received at least three doses of vaccine in the last six months were a sixth less likely to develop severe COVID-19 illness than those that were never vaccinated (Table 2). Conversely, those that were never vaccinated were six times more likely to experience severe illness than those that were recently vaccinated with at least the third dose. Of particular note is that the number of vaccine doses received had less impact on illness severity risk than the timing of the doses received. Those vaccinated in the last six months with only one to two doses had lower odds of developing severe illness than those vaccinated more than six months ago, irrespective of the number of doses received (Table 2). Having had a prior episode of COVID-19 reduced the risk of severe illness by about a third.

### 3.3. Final Model 

We ranked the risk severity score in the final model into ten percentiles. The risk severity score ‘jumped’ dramatically for the highest percentile group (the 10th percentile) (Figure 1). Patients with a COVID-19 infection at the 10th percentile were labeled as patients at increased risk for severe illness. Correspondingly, only 0.1% of those COVID-19 patients with a risk severity score between the 1st and 9th percentiles had severe COVID-19 illness, compared to 2.8% of the patients who ranked at the 10th percentile (*p* < 0.001).

### 3.4. Model Accuracy

When the model was applied to the ‘test’ population, the risk probability distributions by the score were comparable to those that were described above for the test population. The sensitivity was 75.2% (750/997), specificity was 90.2% (251,307/278,696), with an ROC of 88.7%. However, the high-risk measure had a poor PPV where only 2.7% of all those with a score of ten had severe illness (750/28,139). 

## 4. Discussion

### 4.1. Main Findings

In this retrospective cohort study, we developed a scoring tool to identify COVID-19 patients that are at risk for severe illness during the Omicron wave of COVID-19 infection in Israel. Severe illness was rare in the study population (0.4%). We identified several risk factors for severe illness: increasing age (with a dramatic increase for the aged 75 and over) and immunosuppression status (with immune-suppressed persons being almost five times more at risk of severe illness). Other risk factors included male gender, comorbidities (heart disease, DM, HTN, cancer, CKD, COPD), complicated medical condition, and living in a nursing home. There were two significant protective factors that were identified: vaccination status (not only the number of doses but also how recently they had been received) and a prior episode of COVID-19 illness. The final model defines high-risk patients as patients with scores in the 10th percentile of risk calculation. Sensitivity and specificity were 75.2% and 90.2%, respectively, with an ROC of 88.7%. However, the PPV was low, only 2.7%. 

### 4.2. Strengths and Limitations

The study is based on all members from a large HMO in Israel that provides healthcare coverage for over 2.5 million citizens from all sectors across the country. The MHS database includes comprehensive details of all illness and hospital episodes and comorbidity and mortality data. While these are strengths of the study, applying the findings here to other HMOs, particularly other countries, should be carried out with caution. Earlier studies had developed models when other, more virulent but less infectious variants were predominant. The present study provides a model designed predominantly for those that are infected with Omicron. This variant was prevalent in most countries, so the findings here are prescient. However, the model’s applicability to other variants needs to be tested. 

### 4.3. Interpretation

Several studies have developed scoring tools to estimate severe morbidity and mortality risk. Many of these studies were conducted in secondary care facilities [12]. Such studies aimed to identify patients that are at high risk of severe morbidity/mortality to prioritize patients for vaccination, close monitoring, and treatment, including medication to reduce severe illness [8,9]. Our study is based on community patients and not hospitalized patients. This is important because during the Omicron wave, most patients were treated in the community [13]. 

The model developed had good sensitivity and specificity (75.2% and 90.2%, respectively), with an ROC of 88.7% for severe COVID-19 (hospitalization or death). The high specificity (90.2%) refers to the ability of the model to identify patients without risk for severe infection. This is good for prioritizing which patients do not need close monitoring by their primary care physician and possible medical treatments (such as anti-viral drugs). The tool is less effective in detecting whose who truly have a high risk for severe disease. Thus, more patients will get close monitoring and medical treatments when using this tool. The low prevalence of severe disease can explain the low PPV in the Omicron wave (only 0.4% had severe disease in our study).

Based on the model, scores were calculated for the total MHS population and updated fortnightly. During the peak of this wave, scoring was essential for prioritizing resources when thousands were being infected daily. While having a low PPV meant that outreach based on the model included many who would ultimately not be hospitalized or worse, this was still a far better method of prioritizing care than attempting to provide active outreach to all (as carried out in previous waves of COVID-19). Scoring is currently used to prioritize those patients that would benefit from prophylactic anti-viral treatments.

We identified two protective factors against serious illness: vaccination status and prior history of COVID-19 infection. Many of the risk factors that were identified here, such as age, gender, socioeconomic status, and comorbidities, have also been identified in other models that have been developed [8,9,10,11,14,15,16]. In another HMO-based Israeli study, the number of doses was found to have a protective effect against hospitalization and mortality. In this study, we also looked at the time from the last dose and found that those that were vaccinated more than six months ago were at greater risk of serious illness. This finding emphasizes the importance of timely booster vaccinations found in other studies [17]. Other studies have also found that prior COVID-19 infection is associated with a lower risk of severe, critical, and fatal disease. To be noted, Dhillon et al. reported an opposite association [18] where they suggested that reinfection with COVID-19 has a slightly higher incidence of mechanical ventilation and admission to the ICU. 

We also found that women who were in their fifth or more month of pregnancy were much more likely to be hospitalized (OR: 89). It is possible that some of these hospitalizations may be policy-driven rather than indicating deterioration in health; physicians may not have wanted to take the risk of caring for this population in the community, particularly in the later months of pregnancy.

## 5. Conclusions

We developed a scoring tool to identify patients that are at increased risk for severe COVID-19. This model allowed us to prioritize patients requiring closer follow-up by their PCP/outreach services and those most that were likely to benefit from anti-viral treatment. 

## Figures and Tables

**Figure 1 biomedicines-11-00767-f001:**
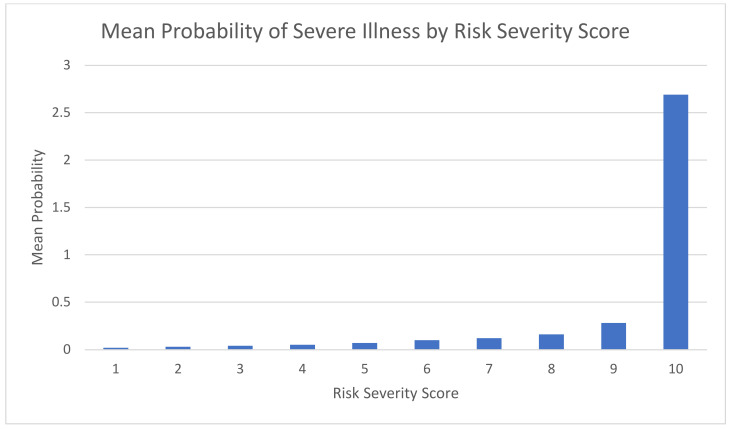
Mean probability by risk severity score, model development population (‘train’ group) Jan 2022 (N = 130,000), Maccabi HealthCare Services, Israel.

**Table 1 biomedicines-11-00767-t001:** Demographic and Health Characteristics of COVID-19 infected population (N = 409,693), 26 November 2021–27 January 2022, Maccabi HealthCare Services, Israel.

Characteristic	Category	n	%
Gender	% Male	184,844	45.1
Age group	<18	140,890	31.4
	18–29	68,545	16.7
	30–44	83,561	20.4
	45–59	77,769	19.0
	60–74	30,921	7.5
	75+	8007	2.0
Socioeconomic Status	Low	69,855	17.1
	Middle	210,251	51.3
	High	129,587	31.6
No. COVID-19 illness episodes	% two or more	31,053	7.6
Vaccine Status	Never vaccinated	110,286	26.9
	Vaccinated 6+ months ago	37,004	9.0
	Received 1–2 doses in last 6 months	55,938	13.7
	Received 3–4 doses in last 6 months	206,465	50.4
Heart disease	% with illness	17,746	4.3
Diabetes	% with illness	17,066	4.2
HTN	% with illness	37,503	9.2
Cancer	% with illness	11,160	2.7
CKD	% with illness	6694	1.6
Immunosuppression status	% with disorder	5215	1.3
COPD	% with illness	2681	0.7
Complicated medical condition	% with illness	1673	0.4
Obesity	% with BMI ≥ 30	51,985	12.7
Pregnancy	% 5+ months pregnant	1886	0.5
Lives in a nursing home	% live in a nursing home	4176	1.0

**Table 2 biomedicines-11-00767-t002:** Factors that were associated with severe illness (hospitalization or mortality) among the COVID-19 infected population (logistic regression model (N = 130,000), 26 November 2021–27 January 2022, Maccabi HealthCare Services, Israel.

Characteristic	Category	N	OR	95% CI
Gender	Female	71,090	1	
	Male	58,910	1.2	1.0–1.5
Age group	<18	44,709	1	
	18–29	21,851	2.7	1.6–4.5
	30–44	26,528	2.2	1.3–3.7
	45–59	24,610	6.6	4.2–10.4
	60–74	9735	16.6	10.4–26.5
	75+	2567	70.4	42.8–115.9
Socioeconomic status	High	41,152	1	
	Middle	66,722	1.6	1.2–2.1
	Low	22,126	1.9	1.4–2.6
No. COVID-19 illness episodes	Once	120,181	1	
	Two or more	9819	0.3	0.2–0.5
Vaccine Status	Never vaccinated	35,076	1	
	Vaccinated 6+ months ago	11,803	0.6	0.4–0.8
	Received 1–2 doses in last 6 months	17,691	0.4	0.2–0.6
	Received 3–4 doses in last 6 months	65,430	0.2	0.1–0.2
Heart disease	No	124,397	1	
	Yes	5603	1.6	1.2–2.0
Diabetes	No	124,629	1	
	Yes	5371	1.4	1.1–1.8
HTN	No	118,147	1	
	Yes	11,853	1.5	1.1–1.9
Cancer	No	126,420	1	
	Yes	3580	1.5	1.1–2.0
CKD	No	127,877	1	
	Yes	2123	1.7	1.2–2.2
Immunosuppression status	No	128,370	1	
	Yes	1630	4.8	3.4–6.7
COPD	No	129,152	1	
	Yes	848	1.9	1.3–2.7
Complicated medical condition	No	129,474	1	
	Yes	526	2.5	1.3–4.6
Obesity	No	113,457	1	
	Yes	16,543	1.0	0.8–1.3
Lives in a nursing home	No	128,673	1	
	Yes	1327	2.5	1.9–3.4
Pregnancy	No/0–4 months pregnant	118,147	1	
	5+ months pregnant	597	82.9	53.0–129.6

## Data Availability

The data presented in this study are available on request from the corresponding author. The data are not publicly available due to ethical considerations.

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
