# Peer review of "Risk Stratification Model for Severe COVID-19 Disease: A Retrospective Cohort Study"

_biomedicines, 2023, doi:10.3390/biomedicines11030767_

Round 1

Reviewer 1 Report

The research develops risk stratification models to identify patients at higher risk of COVID-19 infection and serious illness. For this purpose, the authors designed, developed, and implemented a scoring tool to identify COVID-19 patients at risk for severe illness during the OMICRON wave. This tool was built to assist Maccabi Healthcare Services (MHS) in Israel in allocating resources more effectively. I have the following comments about the research article.

1. The manuscript did not discuss existing works and did not justify the novelty.

2. The manuscript claimed a model is developed. It is not clear what kind of model. Even there is no discussion of the model.  

3. The manuscript discusses some results. But did not discuss experiment environment, parameters, etc.

4. The authors did not discuss data pre-processing and cleaning.

5. How are the results verified?

6. Please discuss the tool more elaborately.

7. Please discuss evaluation metrics.

8.  How is the model’s biasness is handled?

Author Response

  1. The manuscript did not discuss existing works and did not justify the novelty.

We added this to the introduction:

At the beginning of the pandemic, Barda et al. developed a tool for predicting mortality (7). However, this was when no vaccinations existed, and other more virulent variants were predominant. Later, models were introduced to prioritize the vaccination effort (8,9). Hippisley-Cox et al. developed a risk-predicting model for hospitalization or mortality due to COVID-19 (10). They accounted in their model the vaccination status (at this time, only two doses were available). They showed that the absolute risk for hospitalization and mortality was reduced after the second dose. Israel et al. also developed a scoring model including vaccination status. They already included four doses of vaccine (11). They evaluated similar variables as our study did. However, they only addressed the number of vaccines and not their timing and did not consider prior episodes of infection.

The objective of the present study was to design and develop a scoring tool to identify COVID-19 patients at risk for severe illness (hospitalization or mortality). This tool was designed when patients received four vaccine doses and potentially had a prior infection with COVID-19. The novelty of this model is that we considered the number of vaccine doses and their timing. In addition, prior infection was taken into account.

  1. The manuscript claimed a model is developed. It is not clear what kind of model. Even there is no discussion of the model.

In the results we added this paragraph:

Final model

We ranked the risk severity score in the final model into ten percentiles. The risk severity score 'jumped' dramatically for the highest percentile group (the 10th percentile) (Figure 1). Patients with a COVID-19 infection at the 10th percentile were labeled as patients at increased risk for severe illness. Correspondingly, only 0.1% of those COVID-19 patients with a risk severity score between the 1st and 9th percentiles had severe COVID-19 illness, compared to 2.8% of the patients who ranked at the 10th percentile (p<.001).

In the discussion, we added this to the main findings:

The final model defines high-risk patients as patients with scores in the 10th percentile of risk calculation. Sensitivity and specificity were 75.2% and 90.2%, respectively, with a ROC of 88.7%. However, the PPV was low, only 2.7%.

  1. The manuscript discusses some results. But did not discuss experiment environment, parameters, etc.

In the discussion (interpretation), we added:

Our study is based on community patients and not hospitalized patients. This is important because, during the Omicron wave, most patients were treated in the community (13). 

  1. The authors did not discuss data pre-processing and cleaning.

We added to the methods (study design and settings):

MHS' database contains the above mentions parameters for all patients. Therefore there was no need for data pre-processing and cleaning.

  1. How are the results verified?

We added to the methods (study design and settings):

The results were verified by using a 2-step approach. The model was first built on the 'train group' (one-third of the study population) and later tested and verified on the ‘test group' (the remaining two-thirds of the study population). Bias was handled by using multivariate regressions.

  1. Please discuss the tool more elaborately.

We added this to the discussion:

The model developed had good sensitivity and specificity (75.2% and 90.2%, respectively), with a ROC of 88.7% for severe COVID-19 (hospitalization or death). The high specificity (90.2%) refers to the ability of the model to identify patients without risk for severe infection. This is good for prioritizing which patients do not need close monitoring by their primary care physician and possible medical treatments (such as anti-viral drugs). The tool is less effective in detecting who truly has a high risk for severe disease. Thus, more patients will get close monitoring and medical treatments when using this tool. The low prevalence of severe disease can explain the low PPV in the omicron wave (only 0.4% had severe disease in our study). 

  1. Please discuss evaluation metrics.

We added this to the discussion:

The model developed had good sensitivity and specificity (75.2% and 90.2%, respectively), with a ROC of 88.7% for severe COVID-19 (hospitalization or death). The high specificity (90.2%) refers to the ability of the model to identify patients without risk for severe infection. This is good for prioritizing which patients do not need close monitoring by their primary care physician and possible medical treatments (such as anti-viral drugs). The tool is less effective in detecting who truly has a high risk for severe disease. Thus, more patients will get close monitoring and medical treatments when using this tool. The low prevalence of severe disease can explain the low PPV in the omicron wave (only 0.4% had severe disease in our study). 

  1. How is the model’s biasness is handled?

We added this to the methods:

Bias was handled by using multivariate regressions.

Reviewer 2 Report

In this study, a risk stratification model was developed for severe Covid-19 disease.  The design of the study has at least three clear strengths that mark it as potentially highly significant: 1) the wealth of health-related data available on the patients included in the study population; 2) the fact that the Covid-19 cases included in the study are predominantly the result of infection by the same variant of the virus (Omicron) over a very short period of time (< two months); and, 3) the decision to use 1/3 of the population as the train group, upon which the model was developed, and the remaining 2/3 of the cohort as the test group to evaluate the accuracy of the model.  Really, when all is said and done, there is not much that would be considered particularly surprising in the results of the study.  One would expect medical conditions such as diabetes, immunosuppression, pregnancy, etc. to make one more susceptible to a severe Covid-19 outcome.  This is not to say that the study is not informative or worthy of publication.  On the contrary, it provides strong affirmation of protective guidelines that have been set forth almost since the outset of the pandemic and counters the suspicions raised by vaccine opponents.  Indeed, the finding that the timing of vaccination impacts disease severity is important, as it provides clear incentive for keeping one’s vaccination status updated.  For these reasons, the manuscript is considered an important contribution to the field.  Furthermore, it provides hard data to support the notion that the severity of Covid-19 disease is impacted in significant ways by one’s overall health, a direct correlation not to be ignored.

There is only a single criticism concerning a statement in the abstract implying that factors associated with severe disease include vaccination status (patients vaccinated in the previous six months).  This statement is poorly worded and misleading.  As written, it implies that updated vaccine status contributes to severe disease.  Shouldn’t this be changed to patients not vaccinated in the prior six months?

Author Response

There is only a single criticism concerning a statement in the abstract implying that factors associated with severe disease include vaccination status (patients vaccinated in the previous six months).  This statement is poorly worded and misleading.  As written, it implies that updated vaccine status contributes to severe disease.  Shouldn’t this be changed to patients not vaccinated in the prior six months?

Thank you. We changed this statement in the abstract:

Results 409,693 patients were diagnosed with COVID-19 over the two-month study period, of which 0.4% had severe illness. Factors associated with severe disease were age (age>75, OR-70.4, 95% confidence interval [CI] 42.8-115.9), immunosuppression (OR-4.8, 95% CI 3.4-6.7), and pregnancy (5 months or more, OR-82.9, 95% CI 53-129.6). Factors associated with a reduced risk for severe disease were vaccination status (patients vaccinated in the previous six months OR-0.6, 95% CI 0.4-0.8) and a prior episode of COVID-19 (OR-0.3, 95% CI0.2-0.5). According to the model, patients who were in the 10th percentile of the risk severity score were considered at increased risk for severe disease. The model accuracy was 88.7%.

Round 2

Reviewer 1 Report

The manuscript has been revised addressing my comments. It is fine now.